# EGODE: An Event-attended Graph ODE Framework for Modeling Rigid Dynamics

**Jingyang Yuan[1]    Gongbo Sun[2]    Zhiping Xiao[3]\*    Hang Zhou[4]    Xiao Luo[5]\***
**Junyu Luo[1]    Yusheng Zhao[1]    Wei Ju[1]    Ming Zhang[1]\***
[1]School of Computer Science, State Key Laboratory for Multimedia
Information Processing, PKU-Anker LLM Lab, Peking University
[2]University of Wisconsin-Madison [3]University of Washington
[4]University of California, Davis [5]University of California, Los Angeles
yuanjy@pku.edu.cn, gsun43@wisc.edu, patxiao@uw.edu, hgzhou@ucdavis.edu,
xiaoluo@cs.ucla.edu, luojunyu@stu.pku.edu.cn, yusheng.zhao@stu.pku.edu.cn,
juwei@pku.edu.cn, mzhang_cs@pku.edu.cn

## Abstract

This paper studies the problem of rigid dynamics modeling, which has a wide range of applications in robotics, graphics, and mechanical design. The problem is partly solved by graph neural network (GNN) simulators. However, these approaches cannot effectively handle the relationship between intrinsic continuity and instantaneous changes in rigid dynamics. Moreover, they usually neglect hierarchical structures across mesh nodes and objects in systems. In this paper, we propose a novel approach named Event-attend Graph ODE (EGODE) for effective rigid dynamics modeling. In particular, we describe the rigid system using both mesh node representations and object representations. To model continuous dynamics across hierarchical structures, we use a coupled graph ODE framework for the evolution of both types of representations over a long period. In addition, to capture instantaneous changes during the collision, we introduce an event module, which can effectively estimate the occurrence of the collision and update the states of both mesh node and object representations during evolution. Extensive experiments on a range of benchmark datasets validate the superiority of the proposed EGODE compared to various state-of-the-art baselines. The source code can be found at `https://github.com/yuanjypku/EGODE`.

## 1 Introduction

Physics simulations [50, 59] can benefit researchers from many fields by guiding experiments and testing their theories [58]. Among them, simulating rigid collisions has received extensive attention with applications in robotics [19] and graphics [3]. However, high-quality physical simulations usually require complicated computing, which requires extensive computational resources. To solve this issue, data-driven approaches [1, 47] that aim to leverage machine learning for efficient simulators are becoming increasingly popular within the recent years.

In literature, a variety of existing approaches have been proposed to model physical systems [20, 52, 47, 2, 1, 29]. Early attempts usually focus on simulations on regular grids [57, 43, 42, 40] and use convolutional neural networks (CNNs). However, in real-world scenarios, objects are rarely located in regular ways [18, 5, 55, 39]. To increase the applicability, many current works use irregular mesh points [47, 31, 1] to describe objects in physical systems and utilize graph neural networks (GNNs) [16, 28, 38] to capture the interactions between mesh points. In particular, they adopt

---

\*Corresponding authors.

38th Conference on Neural Information Processing Systems (NeurIPS 2024).

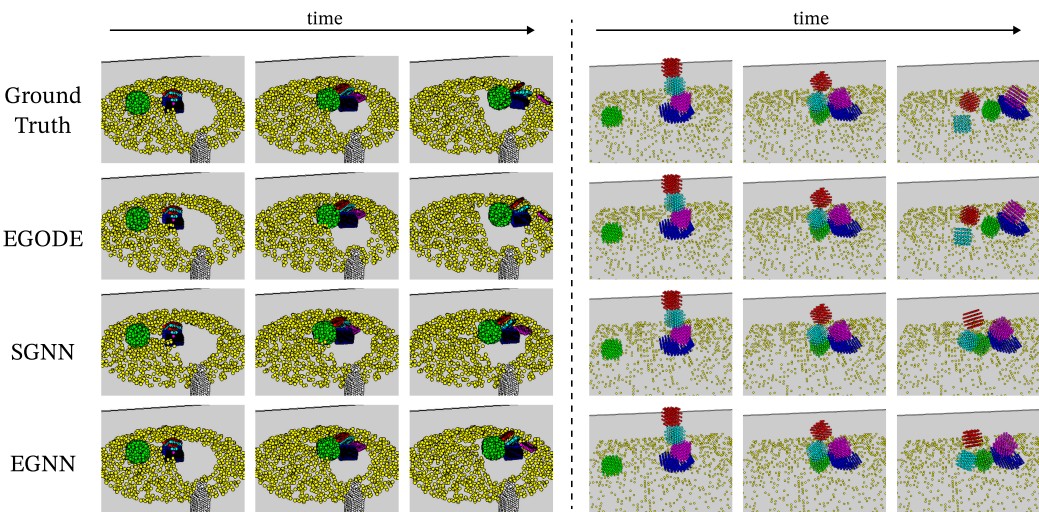

Figure 1: Visualizations of predictions on Physion dataset. EGODE demonstrate the best capability to generate accurate trajectories across diverse scenarios.

an encoder-processor-decoder architecture, which first maps observations of each mesh point at the current time step into the latent space, and then follows the message-passing mechanism [28] to update node representations iteratively. Finally, a decoder is utilized to generate the predicted trajectories.

Despite their notable process, these approaches [20, 52, 47, 2, 1] suffer from three basic obstacles, which could seriously degrade the forecasting performance. **Firstly**, the majority of existing approaches [2, 47] utilize a rollout process to model dynamical systems, which take the predictions at the next time as the input in an autoregressive process. The discrete rollout process makes it difficult to capture the long-term tendency and continuous evolution of complex physical systems. **Secondly**, aside from the continuous evolution, rigid dynamics would face instantaneous changes [20, 54] caused by contact dynamics at certain time steps,whereas the existing approaches often fail to take it into consideration. Therefore, we are required to build a model that can precisely capture the relationships between the dominant continuous evolution and instantaneous changes in complicated rigid-body systems. **Thirdly**, since each object is described by multiple mesh nodes, rigid-body systems intrinsically consist of hierarchical structures across mesh nodes and objects, which increases the difficulty of modeling rigid dynamics.

To address the aforementioned obstacles, in this paper, we propose a novel framework named Event-attend Graph ODE (EGODE) for modeling rigid dynamics. The core idea of our EGODE is to understand the continuous evolution and instantaneous changes in rigid-body systems. To model the hierarchical structures, we introduce both mesh node representations and object representations. To model the continuous dynamics, we adopt its neighbor mesh nodes and the related object to drive the evolution of mesh node representations. Meanwhile, global object representations and the summarized local information jointly determine the evolution of objects. To model the instantaneous changes, we introduce an event module, which estimates the next time when the collision occurs, and then updates both the mesh node representations and the object representations in an iterative manner. Finally, we minimize the standard mean square error (MSE) at both node and object levels. We conduct extensive experiments on a range of benchmark datasets. A comparison of our EGODE with other baselines on the Physion dataset is depicted in Figure 1. The experimental results can validate the superiority of the proposed EGODE over a wide range of competing baselines.

In summary, the contributions of the paper are three-fold: (1) We provide a new perspective of modeling both continuous evolution and instantaneous changes to study rigid dynamics. To the best of our knowledge, we make the first attempt using graph ODE to simulate rigid-body systems. (2) Our EGODE not only utilizes a coupled graph ODE to jointly model the continuous evolution of both mesh nodes' representations and objects' representations, but also introduces an event module to estimate the collision times for instantaneous updating. (3) Comprehensive experiments including

quantitative comparison and visualization on different benchmark datasets demonstrate the superiority of our proposed EGODE over a wide range of competing approaches.

## 2 Related Work

**Data-driven Physical Simulation.** To facilitate physical simulations in different areas, a wide range of researchers leverage machine learning to build effective data-driven simulators [47, 54, 52]. Early attempts usually adopt convolutional neural networks (CNNs) to model physical systems with grid structures [46]. To increase the flexibility of simulators, recent efforts have focused on building simulators on irregular grids [47, 20, 52], which usually leverage graph neural networks (GNNs) [28] to model the interaction between objects. For example, MeshGraphNet [47] adopts an encoder-processor-decoder architecture to predict the next states for effective mesh-based simulations. EGNN [20] considers the subequivariance of physical systems during the message passing process. GNS [52] have validated the potential of graph neural networks for modeling rigid dynamics. However, these approaches cannot handle intrinsic continuity and discontinuity in rigid models while our EGODE is the first work to introduce graph ODE for effective rigid dynamical modeling.

**Graph Neural Networks.** Graph neural networks (GNNs) [28, 26, 25] have been shown efficient in a wide range of vision tasks including cross-modal learning [67, 60], object detection [33, 56] and transfer learning [15, 17, 66, 34, 35]. These approaches usually follow the paradigm of message passing [65], which updates the central nodes by aggregating their neighborhood information iteratively. Through this process, GNNs can learn from geometric structures for downstream tasks. By combining GNNs with neural ODEs [9], a range of continuous GNNs [48, 64, 49, 61] have been developed, which model the neighborhood aggregation in a continuous way. For example, GDERec [49] combines neural ODE with an attention-based GNN to model the interaction signals in recommender systems. However, these approaches usually neglect the instantaneous change in interacting dynamical systems [9]. To handle this, we propose a new continuous GNN framework named EGODE, which can model the instantaneous updating in rigid-body systems.

**Neural Ordinary Differential Equations (ODE).** Compared with classic deep neural networks, neural ODEs [9] aim to include continuous layers rather than discrete ones with extensive applications [44, 7, 63, 4, 21, 8]. The updating rule of neural ODE is accelerated by incorporating adjoint functions with neural ODE solvers [12]. Recently, a range of approaches [10, 14, 41] have been proposed to improve the effectiveness of neural ODE, including augmenting the dimension [10] and regularization terms [14, 41]. Neural ODEs have been adopted to model multi-agent dynamical systems [23, 24, 37, 36], which can deal with irregularly sampled data and partial observations. In this paper, we propose a novel neural ODE framework EGODE, which can model both instantaneous updating and continuous evolution in rigid-body systems.

## 3 Methodology

### 3.1 Problem Definition

We assume that a rigid-body physical system consists of $M$ objects with $N$ mesh points. The state information of each mesh node $i$, $1 \leq i \leq N$, includes the observation vectors (i.e., the position vector $\boldsymbol{x}_i^t$ and the velocity vector $\boldsymbol{v}_i^t$ at time t) and static vectors $\boldsymbol{s}_i$ (e.g., friction of the floor) unrelated to the geometric context. The graph structure is constructed based on positions of mesh nodes, i.e., $\mathcal{G}^t = \{V, E^t\}$ where $V$ collects all the mesh points and $E^t$ consists of all the edges at the time step $t$. Following previous works [20], we build an edge when the distance between two mesh points is below a given threshold, making up the edge set $E^t$. Given the initial states $\mathcal{G}^0$, we aim to predict the future trajectories $\boldsymbol{X}^{1:T}$ where $\boldsymbol{X}^t$ denotes the position matrix at the time step $t$.

### 3.2 Framework Overview

In this work, we study the problem of modeling rigid-body physical systems, which is challenging due to the hierarchical evolution of systems and instantaneous changes from collisions. Towards this end, we introduce a new framework named EGODE, which models the evolution of physical systems in a continuous manner with the consideration of instantaneous events. In particular, EGODE first drives the dynamics of mesh nodes using both its surrounding mesh nodes and the associated object.

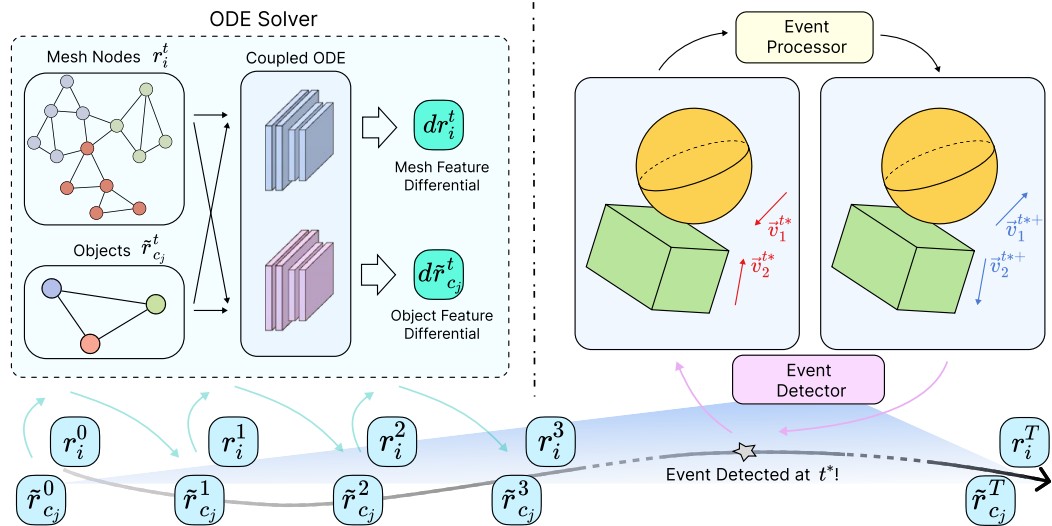

Figure 2: Overview of EGODE. We employ a coupled graph ODE framework with an event module for collision modeling. The coupled graph ODE structure naturally captures the continuous dynamics inherent in interacting systems. $r_i^t$ and $\tilde{r}_{c_j}^t$ represents feature of mesh node and object node at time $t$, respectively. Complementarily, the event module is designed to effectively handle potential instantaneous changes, such as those arising from collisions.

Moreover, the object-level dynamics is driven by local node information and global system states. To effectively capture the instantaneous collision, we introduce a learnable event module, which detects the potential collision time and updates the instantaneous change iteratively during ODE evolution. An overview of the proposed EGODE can be found in Figure 2 and the details are introduced below.

### 3.3 Coupled Graph ODE

Previous works [20, 52, 47] usually adopt graph neural networks (GNNs) [16] to predict the states of objects at the next step, followed by autoregressive iterations for long-term predictions. However, these approaches cannot capture the long-term tendency and continuous evolution in physical systems [23]. Towards this time, we introduce a graph ODE framework, which can capture continuous dynamics in interacting systems naturally. Moreover, since we have both objects and mesh nodes in rigid-body systems, our ODE framework consists of a coupled architecture, which models hierarchical structures in a unified way [24, 68, 20].

In particular, we first initialize each latent state using the static vector, and then concatenate all the dynamical vectors with it for each node into a new vector $r_i^t$ as follows:

$$r_i^t = [x_i^t, v_i^t, h_i^t], \tag{1}$$

where $h_i^t$ denotes the corresponding hidden state with $h_i^0 = s_i$. To model the continuous evolution, we introduce a neural graph ODE framework by combining neural ODE [9] with GNNs. Our graph ODE drives the dynamics of the system using the interaction between each node and its neighbors. Formally, we have:

$$\frac{dr_i^t}{dt} = \phi^l(\sum_{j \in \mathcal{N}^t(i)} \psi^l(r_i^t, r_j^t)), \tag{2}$$

where $\mathcal{N}^t(i)$ collects the neighbors of mesh node $i$ at the time step $t$. $\psi^l(\cdot, \cdot)$ aims to capture the interaction between each object and its neighbors and $\phi^l(\cdot)$ produces the summarized influence from the neighborhood to drive the evolution of the system. Moreover, in rigid-body systems, there are naturally hierarchical structures ranging between mesh nodes and objects. To model the hierarchy effectively, we introduce the states at the object level, by calculating the average of their corresponding observation vectors. In formulation, the object-level vectors can be initialized as follows:

$$\tilde{x}_{c_j}^0 = \frac{1}{|i : o(i) = j|} \sum_{i:o(i)=j} x_i^0, \tag{3}$$

$$\tilde{\boldsymbol{v}}_{c_j}^0 = \frac{1}{|i:o(i)=j|} \sum_{i:o(i)=j} \boldsymbol{v}_i^0, \tag{4}$$

where $o(i)$ returns the object $j$ corresponding to the mesh point $i$. We also utilize $\tilde{\boldsymbol{h}}_{c_j}^t$ to denote the latent object representation of each object and have $\tilde{\boldsymbol{r}}_{c_j}^t = [\tilde{\boldsymbol{x}}_{c_j}^t, \tilde{\boldsymbol{v}}_{c_j}^t, \tilde{\boldsymbol{h}}_{c_j}^t]$, which is then incorporated into the evolution of all its corresponding nodes. In other words, we re-write Equation 2 into:

$$\frac{d\boldsymbol{r}_i^t}{dt} = \phi^l([\sum_{i' \in \mathcal{N}^t(i)} \psi^l(\boldsymbol{r}_i^t, \boldsymbol{r}_{i'}^t), \tilde{\boldsymbol{r}}_{c_j}^t]), \tag{5}$$

where $\tilde{\boldsymbol{r}}_{c_j}^t$ can provide high-level semantics for dynamics modeling. To obtain $\tilde{\boldsymbol{r}}_{c_j}^t$, we include another graph ODE to drive the evolution at the object level. Here, we not only connect each object with all the other objects in the system for global understanding, but also learn from its corresponding mesh node for local information. In formulation, we have:

$$\frac{d\tilde{\boldsymbol{r}}_{c_j}^t}{dt} = \phi^g([\sum_{j=1}^{M} \psi^g(\tilde{\boldsymbol{r}}_{c_j}^t, \tilde{\boldsymbol{r}}_{c_{j'}}^t), \frac{1}{|i:o(i)=j|} \sum_{i:o(i)=j} \boldsymbol{r}_i^t]), \tag{6}$$

where $\phi^g(\cdot)$ and $\psi^g(\cdot, \cdot)$ are two learnable functions to object-level updating with different parameters, $M$ is the number of objects. In the right hand, the first term calculates the interaction between different objects and the second term summarizes the states of its associated node presentations. In the end, we combine both Equation 5 and Equation 6 to jointly solve the coupled ODE, which can not only model the continuous evolution in physical systems, but also output the trajectory at any time step. The whole coupled ODE can be solved by traditional neural ODE solver [9].

### 3.4 Event Module for Collision Modeling

We have introduced a graph ODE framework to model the continuous evolution in physical systems. However, rigid-body dynamical systems [20, 1, 2] could include instantaneous change during the collision between objects. In this case, our coupled graph ODE could be incapable of sufficiently modeling these discontinuous systems. To tackle this, we include a learnable event module to estimate the time of potential collision, which can guide the adjustment to the states of different mesh points and objects [8, 51, 47].

One basic solution to model the event (i.e., collision) occurrence is to utilize prior knowledge (e.g., shapes of objects) as well as position information, which could be unavailable in real-world applications. As a consequence, to make our data-driven model more generalized, we utilize a learnable event function condition on the pairwise states of mesh points, which can be formalized as $g(t, [\boldsymbol{x}_i^t, \boldsymbol{v}_i^t, \boldsymbol{x}_{i'}^t, \boldsymbol{v}_{i'}^t])$. This event function is capable of continuously detecting the time when the collision between mesh nodes $i$ and $i'$ occurs using the following equation:

$$g(t, [\boldsymbol{x}_i^t, \boldsymbol{v}_i^t, \boldsymbol{x}_{i'}^t, \boldsymbol{v}_{i'}^t]) = 0. \tag{7}$$

Note that event fucntion is only calculated between point-pairs whose distance is within a threshold to avoid square complexity. After solving Equation 7 using the neural ODE solver, we can obtain the collision time $t^*$. Note that in rigid-body systems, the collisions of mesh nodes from different objects would bring in instantaneous change on all the mesh nodes in their related objects. Here, we update the states of observations using the current states and the object that it collides with. In formulation, the vector after the collision can be written as:

$$\boldsymbol{r}_i^{t*+} = \phi^{l*}([\sum_{j \in \mathcal{C}^{t*}(i)} \psi^{l*}(\boldsymbol{r}_i^{t*}, \boldsymbol{r}_j^{t*}), \tilde{\boldsymbol{r}}_{c_j}^{t*}]). \tag{8}$$

Here, $\mathcal{C}^{t*}(i)$ collects all the mesh nodes belonging to the object that it collides with, and $\phi^{l*}(\cdot)$ and $\psi^{l*}(\cdot)$ are two new learnable functions for instantaneous updating. Through this, we involve an immediate updating at the time step $t^*$ from $\boldsymbol{r}_i^{t*}$ to $\boldsymbol{r}_i^{t*+}$, which can simulate the collision between different objects. Similarly, we can update the state of each object as:

$$\tilde{\boldsymbol{r}}_{c_j}^{t*+} = \phi^{g*}([\psi^{g*}(\tilde{\boldsymbol{r}}_{c_j}^{t*}, \tilde{\boldsymbol{r}}_{c_{j'}}^{t*}), \frac{1}{|i:o(i)=j|} \sum_{i:o(i)=j} \boldsymbol{r}_i^{t*}]), \tag{9}$$

---

**Algorithm 1** Updating Algorithm of EGODE

---

**Require:** Observation data, Future ground truth
**Ensure:** Graph ODE framework;
 1: Initialize the vectors $\boldsymbol{r}_i^0$ and $\tilde{\boldsymbol{r}}_i^0$;
 2: **repeat**
 3:    Forwarding our coupled graph ODE Equation 5 and Equation 6;
 4:    Solve Equation 7 to estimate the collision time during the iterative computation of the ODE;
 5:    Update both node representations and object representations using Equation 8 and Equation 9;
 6: **until** $t > T$
 7: Calculate the loss objective in Equation 12;
 8: Update the parameters of graph ODE framework;

---

where $j'$ denotes the object to have the collision with $j$, and $\phi^{g*}(\cdot)$ and $\psi^{g*}(\cdot)$ are for object-level instantaneous updating. The first term calculates the collision between two objects and the second term models the average of updated mesh node representations. Finally, the whole event-attended graph ODE can be solved by iteratively calculating the next collision using graph ODE and updating the corresponding state with Equation 8 and Equation 9. In this way, we integrate the instantaneous updating into the ODE-based continuous evolution to model the rigid dynamics.

### 3.5 Training Objective

To optimize our graph ODE framework, we first output the observation at different time steps and then minimize the standard mean square error (MSE) loss between the predicted trajectories $\hat{\boldsymbol{X}}^t$ and the ground truth $\boldsymbol{X}^t$:

$$\mathcal{L}^l = \sum_{t=T_0+1}^{T} ||\hat{\boldsymbol{X}}^t - \boldsymbol{X}^t||_2^2. \tag{10}$$

Moreover, we minimize the MSE loss at the object level as:

$$\mathcal{L}^g = \sum_{t=T_0+1}^{T} ||\hat{\tilde{\boldsymbol{X}}}_c^t - \tilde{\boldsymbol{X}}_c^t||_2^2, \tag{11}$$

where $\hat{\tilde{\boldsymbol{X}}}_c^t$ is the predicted object-level matrix and $\tilde{\boldsymbol{X}}_c^t$ denotes the ground truth. Finally, we combine both Equation 10 and Equation 11 as:

$$\mathcal{L} = \mathcal{L}^l + \lambda \mathcal{L}^g, \tag{12}$$

where $\lambda$ is a parameter to balance the losses. The whole updating algorithm can be summarized in Algorithm 1.

## 4 Experiments

### 4.1 Experimental Settings

***Datasets.*** Our proposed model EGODE is evaluated on two physical dynamics datasets, i.e., Rigid-Fall [30] and Physion [6]. RigidFall simulates collisions and interactions between three rigid cubes during falling under a varying gravitational acceleration. Physion, a large-scale dataset and benchmark for physical system interaction evaluation, models both rigid and soft-body collisions for 8 distinct scenarios, which are Dominoes, Contain, Collide, Drop, Roll, Link, Support, and Drape. Each scenario consists of 2000 training trajectories and 150 testing simulations. The two datasets both treat objects as assemblies of particles.

***Baselines and metrics.*** We compare the performance of EGODE with a range of classical methods and state-of-art methods including MLP, RNN, SocialODE [62], GNS [52], DPI-Net [30], EGNN [53], GMN [22], SGNN [20], and SEGNO [32]. We utilize two evaluation metrics: i.e., Contact prediction accuracy and Mean Square Error (MSE).

***Implementation Details.*** We implement our baseline models using Pytorch [45] and torchdiffeq [27]. We adopt the same hyper-parameters and training strategy for both Physion and RigidFall datasets as

Table 1: Results of compared methods on Physion (Accuracy%). **Bold** numbers highlight the best performance and ± represents the standard deviation.

| Methods | Dominoes | Contain | Link | Drape | Support | Drop | Collide | Roll |
|---|---|---|---|---|---|---|---|---|
| MLP | 52.5±0.4 | 58.2±1.0 | 52.3±0.7 | 51.9±0.2 | 54.2±0.8 | 52.5±0.4 | 65.4±1.1 | 72.8±0.5 |
| RNN | 52.2±0.5 | 57.4±1.1 | 53.1±0.9 | 52.3±0.3 | 53.8±0.6 | 53.4±0.4 | 68.4±1.0 | 74.5±0.4 |
| SocialODE | 53.2±0.7 | 59.1±1.8 | 54.2±1.0 | 53.5±0.5 | 55.8±0.8 | 53.1±0.8 | 69.2±0.9 | 75.3±0.4 |
| GNS | 74.8±1.5 | 72.6±1.5 | 61.0±1.7 | 57.4±1.3 | 64.8±1.6 | 63.7±0.7 | 83.8±0.7 | 78.2±2.5 |
| DPI | 70.6±0.7 | 70.7±1.6 | 66.3±2.6 | 52.1±1.1 | 65.6±0.4 | 72.8±0.5 | 82.2±1.8 | 79.9±0.6 |
| EGNN | 70.8±1.6 | 67.5±1.7 | 59.2±2.0 | 54.1±1.4 | 55.3±1.6 | 69.3±1.8 | 79.7±0.4 | 80.9±0.8 |
| GMN | 54.6±1.0 | 66.6±1.2 | 50.7±4.5 | 59.0±2.9 | 61.8±2.4 | 56.2±1.6 | 81.0±0.7 | 80.2±0.9 |
| SGNN | 88.8±2.0 | 78.3±1.3 | 72.6±1.0 | 60.6±0.5 | 71.4±1.3 | 73.9±1.1 | 83.0±1.5 | 84.2±0.7 |
| SEGNO | 88.2±1.6 | 76.2±1.5 | 73.5±1.3 | 59.4±0.7 | 68.3±1.2 | 72.8±0.9 | 85.5±1.2 | 82.4±0.6 |
| EGODE | **94.7±1.4** | **79.0±1.3** | **75.0±1.1** | **61.7±0.6** | **71.7±0.8** | **75.3±1.3** | **90.0±1.0** | **85.7±0.8** |

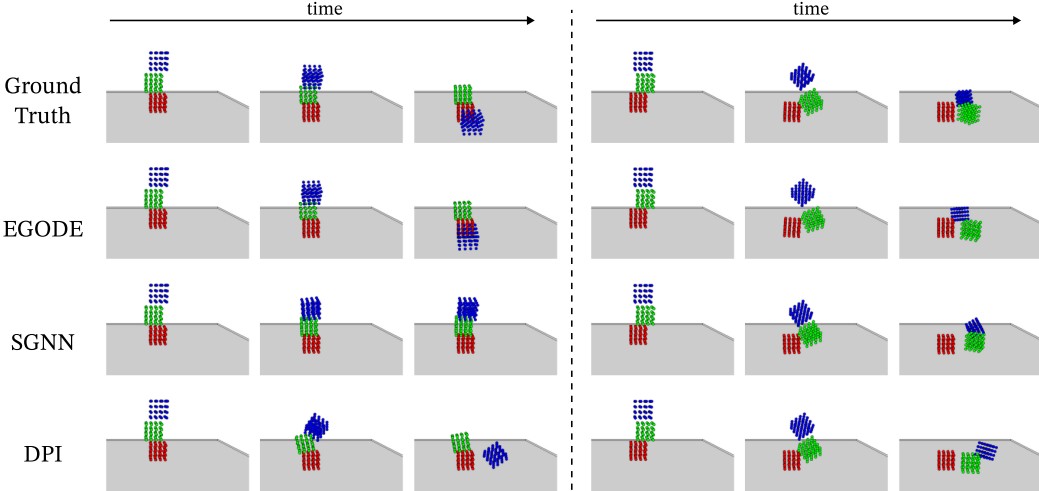

Figure 3: Visualizations of predictions on the RigidFall Dataset. EGODE demonstrates the best capability to generate accurate trajectories.

mentioned by SGNN [20]. In our method, we initialize all MLP layers with a hidden size of 200. An Adam optimizer with the initial learning rate of 0.0001 is adopted during training. We also employ an early stopping strategy of 10 epochs according to validation loss. The batch size is set to 1 for Physion and 8 for RigidFall dataset. To ensure a fair comparison, we initialize all baseline models' parameters based on corresponding papers and then fine-tune them to achieve the best results.

## 4.2 Performance Comparison

The comparison results for Physion and RigidFall datasets are shown in Table 1 and Table 2 accordingly. For the RigidFall dataset, we follow the comparison strategy of SGNN to assess the model's performance with different sizes of the training dataset. From the results, we have three observations. *Firstly*, although the best baseline varies between the datasets, it is generally observed that hierarchical models outperform particle-level models, which confirms that hierarchical methods inherently capture the intrinsic attributes in rigid-body systems and reduce the difficulty of modeling dynamics. *Secondly*, our EGODE demonstrates highest contact prediction accuracy over all baseline models in Physion dataset. In particular, compared to the best baseline SGNN on Dominoes and Collide scenario, our proposed EGODE achieved an increase in prediction accuracy of 5.9% and 4.5%, respectively. *Thirdly*, we observe our EGODE has more robust prediction results than baseline models in long-period prediction on RigidFall dataset, with just a small-scale training set. We attribute the remarkable performance of the proposed EGODE to three key reasons: (1) Introduction of neural ODE in EGODE, provides superior generalization for dynamic systems, especially in long-term prediction scenarios. The neural ODE allows for more accurate modeling for continuous systems, thereby enhancing the overall performance of our model. (2) Introduction of a coupled architecture.

Table 2: Prediction MSE ($\times 10^{-2}$) of compared methods on RigidFall, **Bold** numbers highlight the best performance.

| Methods | |Train| = 500 | | |Train| = 1000 | | |Train| = 5000 | |
|---|---|---|---|---|---|---|
| | $t = 20$ | $t = 40$ | $t = 20$ | $t = 40$ | $t = 20$ | $t = 40$ |
| MLP | 3.25±1.71 | 8.59±4.54 | 2.05±1.19 | 5.82±3.43 | 1.45±0.90 | 4.15±2.68 |
| RNN | 2.87±1.44 | 7.76±4.04 | 2.00±1.11 | 5.35±3.21 | 1.47±0.87 | 3.68±2.37 |
| SocialODE | 2.32±1.17 | 6.01±3.15 | 1.45±0.84 | 4.13±2.35 | 1.03±0.61 | 2.90±1.87 |
| GNS | 2.21±1.03 | 3.98±2.09 | 1.28±0.42 | 2.88±2.32 | 0.95±0.59 | 2.67±1.06 |
| DPI | 1.62±0.59 | 4.46±2.41 | 0.71±0.58 | 4.03±2.76 | 0.51±0.48 | 2.68±2.36 |
| EGNN | 0.94±0.96 | 2.98±2.60 | 1.18±0.51 | 2.79±0.89 | 0.90±0.47 | 2.84±1.13 |
| GMN | 2.25±1.00 | 5.42±2.81 | 1.65±1.65 | 5.45±1.98 | 1.22±0.76 | 2.65±0.86 |
| SGNN | 0.32±0.35 | 1.07±1.23 | 0.32±0.21 | 0.73±0.85 | 0.19±0.19 | 0.74±1.37 |
| SEGNO | 0.64±0.33 | 2.19±1.15 | 0.60±0.34 | 2.21±1.30 | 0.38±0.22 | 1.44±0.92 |
| EGODE | **0.17±0.10** | **0.71±0.53** | **0.17±0.13** | **0.49±0.42** | **0.12±0.11** | **0.46±0.42** |

Table 3: Comparisons between our EGODE and its variants on Physion.

| Methods | Dominoes | Contain | Link | Drape | Support | Drop | Collide | Roll |
|---|---|---|---|---|---|---|---|---|
| EGODE w/o O | 89.8±1.7 | 78.5±1.2 | 73.5±1.2 | 60.8±0.6 | 69.9±0.7 | 74.1±1.6 | 86.3±1.2 | 84.2±0.8 |
| EGODE w/o C | 90.7±1.0 | 78.5±1.8 | 74.2±0.9 | 61.1±0.8 | 70.4±0.8 | 74.5±1.5 | 88.0±1.1 | 84.7±0.9 |
| EGODE w/o E | 90.3±1.5 | 78.5±1.3 | 74.2±1.1 | 60.9±0.4 | 70.0±1.1 | 74.3±1.7 | 86.9±0.7 | 84.5±0.9 |
| EGODE | **94.7±1.4** | **79.0±1.3** | **75.0±1.1** | **61.7±0.6** | **71.7±0.8** | **75.3±1.3** | **90.0±1.0** | **85.7±0.8** |

Our EGODE incorporates both objects and mesh nodes in the rigid body, enabling effective modeling of the dynamic system. (3) Introduction of an event module for collision modeling helps EGODE effectively tackle complex and diverse instantaneous events in rigid body motion, thereby enhancing the performance across different scenarios.

### 4.3 Ablation Study

We analyze our EGODE and evaluate the model's effectiveness in various aspects. In particular, we introduce three model variants as follows: (1) EGODE *w/o O*, which removes neural ODE; (2) EGODE *w/o C*, which removes coupled architecture; (3) EGODE *w/o E*, which removes event module for collision modeling; The results are presented in Table 3. We observe that removing any of the three components leads to an obvious drop in performance on most datasets and tasks. Notably, EGODE *w/o O* causes the most performance degradation. This indicates the continuous method is insufficient to capture the intricate information inherent in rigid body dynamics. We can also conclude from the experiment results that the coupled architecture and event module are crucial for accurately predicting rigid body systems, by effectively aggregating local information, broadcasting global information, and modeling collision events.

### 4.4 Sensitivity Analysis

In this section, we investigate how the hyperparameters, i.e. $\lambda$ in Equation 12 and the distance threshold $d$ for mesh graph construction. The results shown in Figure 4 indicate that the model achieves optimal performance when $\lambda = 1$ when other parameters are fixed. The experiments also suggest that our EGODE exhibits overall stability and robustness across different $\lambda$. The impact of distance threshold $d$ is also analyzed. When $d$ is relatively small, it reduces the connectivity between mesh nodes within an object, hindering information propagation in the network. Conversely, when $d$ is relatively large, redundant interactions might be introduced between objects, slightly affecting model performance. The optimal value of $d$ is depending on the density of mesh nodes and the scale of the objects. Ultimately, we set $\lambda = 1$ and $d = 0.08$ respectively in our experiments.

### 4.5 Generalization Performance

Since the latent embedding $r_i^t$ contains $x_i^t, v_i^t, h_i^t$, it is evident that the left-hand sides of Equation 5 and Equation 6 naturally encompass the acceleration $dv_i^t/dt$, thereby adhering to the fundamental

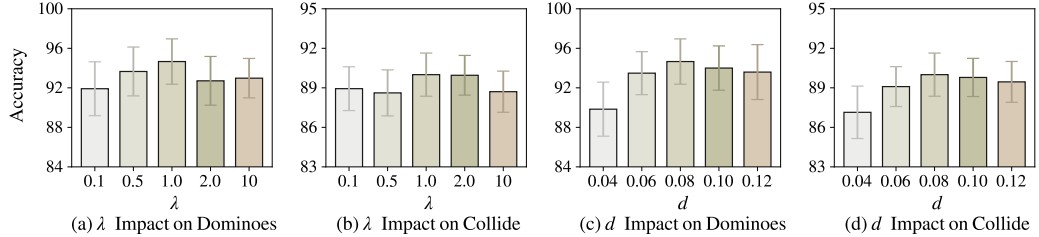

Figure 4: Sensitivity analysis of our EGODE on Dominoes and Collide. The bar charts and error bars describe the accuracy and the 80% confidence intervals, respectively.

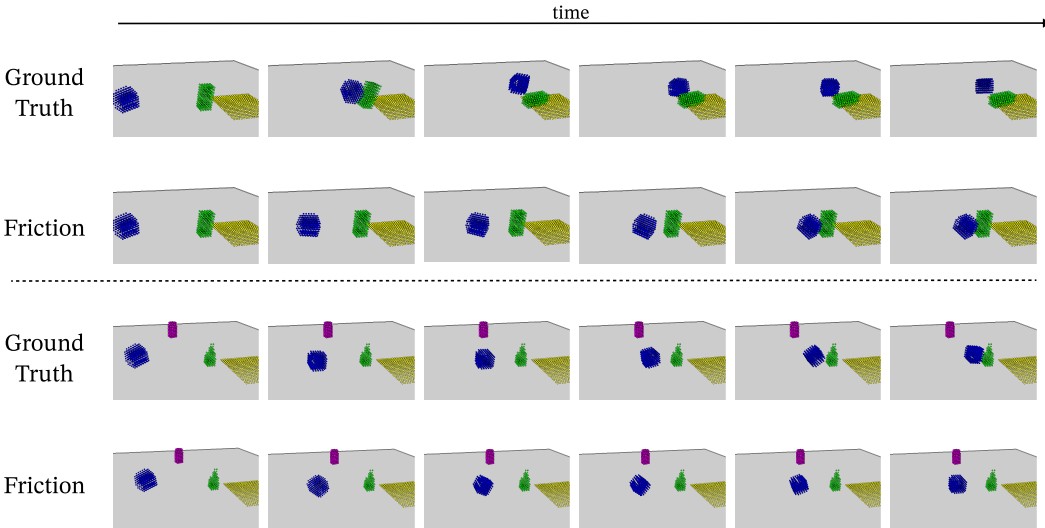

Figure 5: External force field simulation compared with ground truth.

form of Newton's second law. Consequently, by introducing a term representing external forces in Equation 5, we can effectively simulate the presence of additional external forces during the ODE integration process. The external force term in the above setting can be an arbitrary function of position, velocity, and object attribute. Therefore, by defining this term as various force field functions, EGODE can readily simulate the motion of rigid bodies under the influence of different external forces. Detailed formulations about the external force can be found in the Appendix A.

In our experiment, we employ the most common form of resistive force, which is proportional to velocity. We conducte experiments on the Collision scenario from the Physion dataset. As illustrated in Figure 5, when the resistive force is incorporated into the ODE simulation, a notable change in the motion dynamics is observed. In the ground truth, the blue cuboid possesses sufficient energy to collide with and topple the static objects. However, with the introduction of resistive force, the cuboid's velocity and kinetic energy are significantly consumed. These experimental results substantiate the efficacy of our ODE formulation in effectively modeling and transferring motion under the influence of force fields. This remarkable generalization capability stems from the inherent continuity and differentiability properties of our proposed EGODE.

## 5   Conclusion

In this paper, we investigate the problem of rigid dynamics modeling and propose a new approach named EGODE to solve the problem. Our EGODE uses both mesh node representations and object representations to describe the rigid system. More importantly, it adopts a coupled graph ODE architecture to capture the evolution of dynamical systems. To model the occurrence of collisions, EGODE adopts an event module that provides instantaneous updating for the states of mesh node and object representations. Extensive experiments of various benchmark datasets validate the superiority

of EGODE in comparison to different state-of-the-art methods. In future works, we will extend our EGODE to more real-world scenarios including fluid simulation and human trajectory forecasting.

**Broader Impacts and Limitations.** This study introduces an effective data-driven approach EGODE for modeling rigid dynamics, offering a new perspective on collision event modeling in rigid dynamics. One limitation of our work is that our EGODE is unable to accommodate rigid body hinges and deformable objects. Future works will extend EGODE to these more generalization scenarios.

## ACKNOWLEDGEMENTS

This paper is partially supported by the National Key Research and Development Program of China with Grant No. 2023YFC3341203 as well as the National Natural Science Foundation of China with Grant Numbers 62276002 and 62306014.

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

# A  Additional Explanation to Generalization Performance

Since $r_i^t$ contains $x_i^t, v_i^t, h_i^t$, we can effectively simulate the presence of additional external forces during the ODE integration process by introducing a slight modification to Equation 5:

$$\frac{dr_i^t}{dt} = \phi^l([\sum_{i' \in \mathcal{N}^t(i)} \psi^l(r_i^t, r_{i'}^t), \tilde{r}_{c_j}^t]) + [\mathbf{0}, F(x_i^t, v_i^t, h_i^t), \mathbf{0}], \tag{13}$$

where $F$ is an arbitrary function of position, velocity, and object properties, we can deduce from Newton's laws that $F$ corresponds to an additional acceleration term compared to the dynamics described by Equation 5. Consequently, by defining $F$ as various force field functions, EGODE can easily simulate the motion of rigid bodies under the influence of different external forces.

In our experiments, we use the most common form of resistive force, i.e., $F(x_i^t, v_i^t, h_i^t) = -\gamma v_i^t$ where $\gamma$ denotes a constant resistive coefficient. Then the dynamics of $r_i^t$ can be formulated as follows:

$$\frac{dr_i^t}{dt} = \phi^l([\sum_{i' \in \mathcal{N}^t(i)} \psi^l(r_i^t, r_{i'}^t), \tilde{r}_{c_j}^t]) + [\mathbf{0}, -\gamma v_i^t, \mathbf{0}] \tag{14}$$

We can obtain the speed components from both ends of the equation at the same time, and then simplify the formulation:

$$\frac{dv_i^t}{dt} = [\phi^l([\sum_{i' \in \mathcal{N}^t(i)} \psi^l(r_i^t, r_{i'}^t), \tilde{r}_{c_j}^t])]_v - \gamma v_i^t$$

$$\frac{dv_i^t}{dt} + \gamma v_i^t = [\phi^l([\sum_{i' \in \mathcal{N}^t(i)} \psi^l(r_i^t, r_{i'}^t), \tilde{r}_{c_j}^t])]_v \tag{15}$$

$$\frac{d(e^{\gamma t} v_i^t)}{dt} = e^{\gamma t}[\phi^l([\sum_{i' \in \mathcal{N}^t(i)} \psi^l(r_i^t, r_{i'}^t), \tilde{r}_{c_j}^t])]_v$$

This function about $v_i^t$ is similar to a classic decay differential equation. In particular, when there are tiny interactions between nodes and no collisions between objects, the right hand of Equation 15 approaches to 0, and the solution can be approximated as:

$$v_i^t \approx C_i e^{-\gamma t} \tag{16}$$

where $C_i$ is a constant decided by initial conditions. The expression demonstrates particles whose velocity decays exponentially in space. Although the derivation above is based on various assumptions, our experiments in Section 4.5 showed similar results.

# B  Details of Baselines

Our EGODE is compared with a range of competitive methods including MLP, RNN, GNS [52], DPI-Net [30], EGNN [53], GMN [22], SGNN [20], SocialODE [62], and SEGNO [32]. The details of each method are depicted as follows:

- **MLP**: A classical machine learning method applied to the Rigid-body collision task.

- **RNN**: A classical method for time-series prediction, used to model rigid-body movement and predict subsequent motion steps.

- **GNS** [52]: A discrete method that models physical interactions via particle representation using Graph Neural Networks (GNNs). It consists of an encoder-processor-decoder architecture

- **DPI-Net** [30]: A GNN-based method that learns dynamics via particle representation with different materials, including fluids, gases, soft and rigid objects. It showcases the generalization capability with a learned particle dynamics model in real-world control tasks.

- **EGNN** [53]: A GNN-based method designed for dynamics modelling. The model subjects to several equivariance constraints including rotation, translation, and permutation, complying with physics rules.
- **GMN** [22]: A GNN-based method designed for learning dynamics. In addition to rotation, translation, and permutation constraints, GMN also adds geometric constraints, making it geometrically equivariant for interacting objects in the real world.
- **SGNN** [20]: A discrete method which relaxes the equivariant constraints (rotation/translation/permutation) to subequivariance due to external fields like gravity. It consists of both object-level and particle-level message-passing.
- **SocialODE** [62]: An encoder-decoder based architecture that adopts Neural ODE to model continuous transition states. The encoder is a spatio-temporal transformer that encodes historical information into a latent vector, and a sequence of latent trajectories is generated through an Ordinary Differential Equation (ODE) solver and recovered by the decoder.
- **SEGNO** [32]: A GNN-based continuous equivariant method using Neural ODE to approximate dynamic trajectories. It also incorporates second-order motion information to enhance modeling capacity.

## C  Details of Datasets

We mainly conduct experiments on two datasets: Physion [6] and RigidFall [30]. Physion is a large-scale dataset and benchmark for physical system interaction evaluation designed by ThreeDWorld [13]. It contains 8 realistically simulated scenarios:

- **Dominoes**: Simulation of dominoes being knocked down one after another.
- **Contain**: Simulation involving collisions with concave rigid bodies.
- **Collide**: Simulation of a rigid body crashing into other rigid bodies at a relatively high speed.
- **Drop**: Simulation of a rigid body falling onto other rigid bodies.
- **Roll**: Simulation of a rigid body sliding and rolling.
- **Link**: Simulation of ring-mounted rigid bodies.
- **Support**: Simulation of a stack of rigid bodies being hit.
- **Drape**: Simulation of a lightweight flexible object falling on rigid bodies.

RigidFall simulates collisions and interactions between three rigid cubes where each cube consists of 64 particles. The three cubes are initially placed in a stack in the air and fall under varying gravitational acceleration.

## D  Details of Evaluation Metric

To compare these baseline models, we utilize two evaluation metrics: i.e., Contact prediction accuracy and Mean Square Error (MSE). The Contact prediction metric is provided by Physion [6] and used to evaluate whether two target objects collide or not in the whole trajectory. Notably, for MSE, we compare the Euclidean coordinates difference directly rather than calculating the difference between the normalized actual position and predictions. This approach enables a more precise quantification of positional discrepancies in three-dimensional space.

- Contact prediction accuracy:

$$Acc = \frac{1}{s} \sum_{j=1}^{s} \mathbf{1}(y_j = \hat{y}_j).$$

- Mean Absolute Error (MAE):

$$MAE = \frac{1}{n} \sum_{i=1}^{n} |x_i - \hat{x}_i|.$$

Table 4: Prediction MSE of compared methods on Physion, **Bold** number highlight the best performance

| Methods | Dominoes | Collide | Roll | Drape |
|---------|----------|---------|------|-------|
| SGNN | $0.762_{\pm 0.015}$ | $3.39_{\pm 0.25}$ | $2.32_{\pm 0.15}$ | $31.3_{\pm 1.5}$ |
| SEGNO | $0.774_{\pm 0.014}$ | $3.57_{\pm 0.21}$ | $2.53_{\pm 0.19}$ | $34.3_{\pm 2.7}$ |
| EGODE | $\mathbf{0.725_{\pm 0.012}}$ | $\mathbf{3.04_{\pm 0.17}}$ | $\mathbf{2.15_{\pm 0.14}}$ | $\mathbf{29.4_{\pm 1.2}}$ |

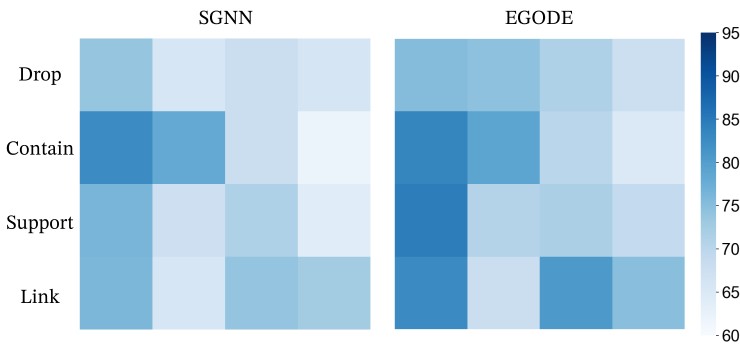

Figure 6: Generalization analysis across different tasks. Row/column records the training or testing phase, respectively. EGODE outperforms the best baseline.

In the above expressions, $y_j$ and $x_i$ represent the ground truth value, while $\hat{y}_j$ and $\hat{x}_i$ represent the predicted value.

# E    Additional Implementation Details

We use torch-geometric [11] and torchdiffeq [27] to complete our code. To enable a fair comparison with previous baselines, our model is only given the initial states of the trajectory $\boldsymbol{X}^0$ to predict future trajectories $\boldsymbol{X}^{1:T}$ for both Physion and RigidFall datasets. An Adam optimizer with the initial learning rate of 0.0001, beta(0.9, 0.999) is adopted during training. A factor of 0.8 and patience of 3 is adopted for the Plateau scheduler. We train our model for 1000 epochs and an early stopping strategy of 10 epochs according to validation loss. To solve the ODE function, we adopt the common Euler ODE solver in our experiment and it performs well in physics modeling task. By adding adaptive collision event module, our model can easily detect collision in the evaluation stage and compute the contact prediction accuracy. For baseline methods without event detector, we follow previous setting of a predefined contact threshold to judge collision and compute corresponding contact accuracy during evaluation. We conduct our experiments on a server with eight NVIDIA A40 GPUs. Since an OpenGL interface and a monitor are required for the visualization process, we visualize our results using a local PC with a single NVIDIA 4090 GPU.

# F    Additional Experiments

## F.1    Prediction MSE on Physion.

We also investigate the prediction MSE in some scenes of the Physion dataset of our method and compared methods. Note that in some scenes (Contain, Drop, Link, Support), objects are often initially positioned in a centrally symmetric manner, therefore all motion patterns centrally symmetric to the ground truth center are reasonable. In this case, MSE is unable to provide an accurate representation of the predictive performance of the model. The results are demonstrated in Table 4. We can indicate that our EGODE outperforms the two strong baselines in all scenes.

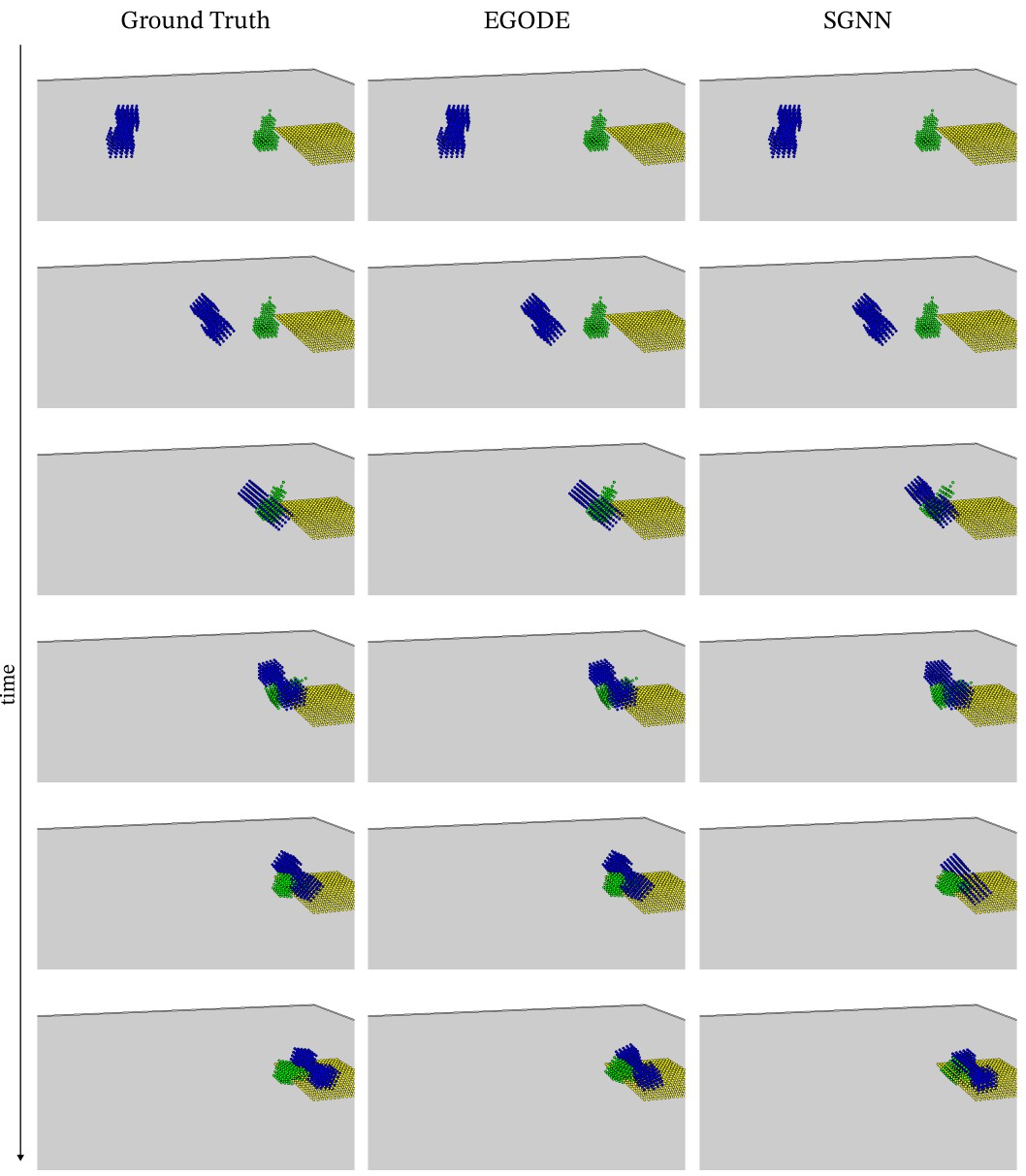

Figure 7: Additional visualizations of predictions on the Physion Dataset.

## F.2   More Generalization Performance

To rigorously evaluate the generalization capability of the proposed EGODE in diverse scenarios, we employ models trained on one specific scenario and test their performance on different scenarios from the Physion dataset. The results illustrated in Fig 6 demonstrate that our proposed method exhibits significantly stronger generalization performance compared to the best baseline, SGNN. This observation highlights the model's proficiency in learning and effectively transferring the underlying principles governing object dynamics and interactions, transcending the specifics of the training scenario. Such superior generalization capability is a testament to the model's ability to capture the intrinsic pattern of rigid dynamics, enabling accurate predictions across diverse scenarios without the need for explicit retraining.

### F.3   More Visualization

As shown in the figure, the additional visualizations of predictions on the Physion Dataset showcase that EGODE outperforms the best baseline SGNN to generate accurate trajectories. EGODE yields predictions closer to the ground truth compared to SGNN. In addition, note that the tip of the green object overlaps with the yellow object in the prediction of SGNN, while our model is more consistent with rigid physical laws. The visualization indicates that EGODE performs better in incorporating physics-based constraints and producing more physically plausible and accurate predictions.

