# OpenReview forum: "EGODE: An Event-attended Graph ODE Framework for Modeling Rigid Dynamics"
_NeurIPS.cc/2024/Conference — NeurIPS 2024 poster_

### Official Review · Reviewer_KaXG · 2024-07-10

**Soundness:** 2
**Presentation:** 2
**Contribution:** 2
**Rating:** 5
**Confidence:** 3

**Summary:**

This paper presents a graph-ODE simulator for capturing rigid body dynamics with collision events. It introduces a design based on object-level and mesh-level representations. It also introduces an event module to capture collision and incorporate it into the network. The paper conducts an evaluation on two rigid-body datasets Rigid-Fall and Physion and compares its performance with multiple learning-based simulators.

**Strengths:**

The paper picks an interesting and potentially impactful research topic. Rigid-body simulation with contact and collisions has wide applications in ML/Robotics, perhaps much more than simulations of any other physics systems (fluids, deformable volumes, cloths, elastic rods, etc.). One of the most critical technical components in such simulations is contact and collision handling, which is also the source of many interesting behaviors of rigid bodies. Rethinking this problem with learning-based techniques may lead to new opportunities, e.g., ML-friendly rigid-body simulators that can be seamlessly integrated into a modern deep-learning pipeline.

**Weaknesses:**

I will combine my comments on “Weaknesses”, “Questions”, and “Limitations” here.

I am confused by this paper’s view on rigid-body simulation. State-of-the-art numerical simulation of rigid bodies with contact, e.g., Isaac Sim and Mujoco 3, is quite powerful. With efficient contact handling algorithms and GPU acceleration, they can simulate fairly complicated (articulated-)rigid bodies like humanoids and quadrupeds at a speed of several millions of time steps per second (https://github.com/google-deepmind/mujoco/discussions/1101). To me, the Rigid-Fall and Physion examples are trivial and solved problems for modern rigid-body simulators. However, the paper did not include these simulators as baselines and limited its comparison with learning-based simulators only. I struggle to see a strong motivation that can support this (lack of) comparison.

The paper also contains several confusing statements regarding physics simulation in its introduction, which I feel are bending the storytelling in the paper’s favor:
1. Paragraph 1: Each sentence alone is technically correct, but combining them gives readers the impression that simulating rigid collision is computationally expensive and data-driven approaches are promising. I am not sure this is the right impression to make. Simulating fluids and deformable bodies is potentially expensive, but simulating rigid bodies with collisions is quite cheap using modern algorithms and hardware.
2. Paragraphs 2-3: Following paragraph 1, these paragraphs lead readers to focus on GNNs (plus standard message passing) and their difficulties in rigid-body simulation. These difficulties do not have to exist in the first place. To me, using graphs to capture rigid-body dynamics is a somewhat contrived idea, because each rigid body contains a small and constant number (6) of DoFs regardless of its mesh resolution. Storing DoFs like x and v on mesh nodes is highly inefficient and unnecessary because they are governed by the 6 DoFs only. GNNs are more suitable for capturing fluids and deformable bodies because their governing equations involve PDEs with spatial derivatives (so information exchange between neighbors is needed) and high DoFs after discretization. In this sense, choosing GNS and DPI as baselines is also contrived: They are not designed specifically for rigid bodies, and they are capable of capturing much more complicated dynamics. Instead, I feel that classic numerical simulators should have been a baseline.

I also have a few more comments regarding the technical method and experiments:
1. It looks like the positions and velocities of all mesh nodes are included in the state variable and evolved in the ODE (Eqns. 1-2). The number of these variables grows as mesh resolution increases, but they are essentially governed by their underlying rigid-body DoFs (6 per rigid body) only. Is it necessary to assign and evolve DoFs at each mesh node?
2. Eqns. 3-4 roughly captures the linear motion of the “center of mass” at each rigid body. Using them as the object-level state does not seem to capture the angular motions and angular velocities of each object. Missing angular information on the object level is a bit counter-intuitive from a physics perspective.
3. The collision events visualized in the figures seem between parametric surfaces (cubes and spheres) only. Such collision events could be easily resolved with closed-form solutions. These examples do not seem to show the benefits of having a mesh representation in collision detection. A more complicated scene with multiple organic surfaces would be a better example to necessitate these meshes.

**Questions:**

See Weakness.

**Limitations:**

See Weakness.

---

> ### Author Rebuttal · Authors · 2024-08-07
>
> We are truly grateful for the time you have taken to review our paper and your insightful review. Here we address your comments in the following.
> > Q1. The Rigid-Fall and Physion examples are trivial and solved problems for modern rigid-body simulators. However, the paper did not include these simulators as baselines and limited its comparison with learning-based simulators only. I struggle to see a strong motivation that can support this (lack of) comparison.
> The paper also contains several confusing statements regarding physics simulation in its introduction, which I feel are bending the storytelling in the paper’s favor.
>
> A1. Although modern numerical simulations can effectively solve rigid body dynamics problems, learn-based methods also possess unique research value. It is based on long-standing considerations followed:
>
> 1. Numerical simulations are not easily integrated with learn-based neural network methods. Specifically, when simulating complex dynamic systems involving both rigid bodies and fluids, it is challenging to delegate fluid simulation and rigid body simulation separately to neural networks and numerical simulators. This makes it difficult to leverage the advantages of neural networks in fluid simulation. On the other hand, end-to-end neural networks can adapt to such complex scenarios by incorporating the simulation of rigid bodies, fluids, and their interactions together (which we are currently exploring). Although this paper does not address fluid simulation, the exploration of learn-based simulations of rigid bodies constitutes a significant precursor to the complex system with rigid and non-rigid. I believe this is one of the most impressive directions for practice and development of our EGODE.
>
> 2. Besides practical implications, our EGODE also has methodological development. We adopt an event module with coupled ODE architecture to model the instantaneous updating of object states. This novel approach can be used not only in rigid body collision simulations but in any dynamic processes involving impulse or angular impulse, and even more broadly in systems including instantaneous state changes (eg. crushing and deformation).
>
> 3. Based on the aforementioned reasons, we have sufficient motivation to study and improve learn-based methods for simulating rigid body dynamics and leverage their strengths in appropriate scenarios, so we only utilized learn-based methods as our baselines for performance comparison.
>
> 4. Moreover, the simulation speed of EGODE is not inferior to numerical simulations. You mentioned that numeral simulators can compute fairly complicated rigid bodies at a speed of several millions of time steps per second. Similarly, our EGODE can also achieve efficient inference by leveraging GPUs and deep learning toolkits such as torch_geometric. In our paper, each scenario visualization contains thousands of nodes and spans 0.5~1 minutes, yet it only requires less than 1 second to simulate and render on a single laptop.
>
> > Q2. It looks like the positions and velocities of all mesh nodes are included in the state variable and evolved in the ODE (Eqns. 1-2). The number of these variables grows as mesh resolution increases, but they are essentially governed by their underlying rigid-body DoFs (6 per rigid body) only. Is it necessary to assign and evolve DoFs at each mesh node?
>
> A2. The positions and velocities of mesh nodes are essential for modeling the interaction and inner-action of objects (eg. contacting, sliding, and pressing), initiating collision events, and performing dynamics calculations. Although the number of these variables increases with higher grid point density, it represents a trade-off between computational cost and performance. Furthermore, the degrees of freedom of surface and interior mesh nodes enable more refined modeling of dynamic phenomena such as compression, friction, and adhesion. Therefore, unless there is a complete absence of interaction between objects, recording the states of these nodes is beneficial.
>
> > Q3. Eqns. 3-4 roughly captures the linear motion of the “center of mass” at each rigid body. Using them as the object-level state does not seem to capture the angular motions and angular velocities of each object. Missing angular information on the object level is a bit counter-intuitive from a physics perspective.
>
> A3. In Eqns. 3-4, linear motion is explicitly recorded at the object nodes of rigid bodies because $x$ and $v$ directly influences the state updating of simulation ($x_i^{t+1}$ is largely determined by $x_i^{t}$and $v_i^{t}$). Although angular momentum and angular velocity are also important, they can be implicitly embedded with other valuable information in hidden state $h^t$, rather than explicitly represented. The model learns to express and utilize them in an optimal way.
>
> > Q4. The collision events visualized in the figures seem between parametric surfaces (cubes and spheres) only. Such collision events could be easily resolved with closed-form solutions. These examples do not seem to show the benefits of having a mesh representation in collision detection. A more complicated scene with multiple organic surfaces would be a better example to necessitate these meshes.
>
> A4. More visualization with more complex objects can be found in Figure.1 and Figure. 7. Specifically, Figure 1 showcases complex objects such as tower-shaped and torus-shaped objects. Our EGODE is designed compatible with rigid bodies of arbitrary shapes, not limited to parametric surface objects. Moreover, our baseline Physion dataset includes objects with various geometries and trajectories of various angles of collision and contact, to validate the generalization capability of our proposed EGODE. Additionally, the results of a complex scenario like Drape, which models interactions between deformable objects and rigid bodies, are reported in Table 3. The visualization of complex scenarios like Drape will be added in the revised version.

---

> > ### Comment · Reviewer_KaXG · 2024-08-12
> > **Thank you for the rebuttal**
> >
> > Thank you for answering my questions. The rebuttal partially addressed my concerns with the paper, so I will increase my rating to 5. However, I feel the rebuttal continues to introduce several ungrounded or biased arguments in favor of the paper, particularly in the answer to Q1.
> >
> > I agree with the potential of learning-based simulators in the long term, but this answer overgeneralizes the proposed approach to fluids/impulse/crushing/deformation without sufficient evidence. For example, I don't see how the claim "This novel approach can be used ... in any dynamic processes involving impulse or angular impulse, and even more broadly in systems including instantaneous state changes (eg. crushing and deformation)." can be justified before we see such a crushing/deformation example.
> >
> > I am also unsure of the answer regarding the neural simulator's efficiency. For rigid-body scenes, "each scenario visualization contains thousands of nodes" does not completely reflect the rigid-body simulation's complexity. The number of objects and constraints from joints and the number of collision events better characterize the simulation difficulty. If I understand correctly, the reported statistics indicate that the proposed simulator is around 30x to 60x faster than real time (0.5-1 min / <1s). This is subpar to what modern rigid-body simulators with contact can handle, and I don't think this can be considered "Similarly" to "a speed of several millions of time steps per second."
> >
> > In summary, despite my positive rating now, I recommend that 1) the authors carefully go over the paper and rephrase any biased or ungrounded claims left and 2) optionally, replicate the examples on modern rigid-body simulators, e.g., Mujoco 3 or Isaac Sim, and report its performance for reference. The primary goal is to give readers and follow-up works a sense of the proposed method's position in rigid-body simulation.

---

> > > ### Author Response · Authors · 2024-08-12
> > >
> > > Thank you for your thoughtful and detailed feedback. We deeply appreciate your recognition of our work's potential and your decision to increase the rating. Your insights are invaluable in helping us improve the quality and clarity of our paper.
> > >
> > > We acknowledge your concern regarding the overgeneralization of our approach, especially event function module, to various dynamic processes such as fluids, impulse, crushing, and deformation. Your point is well taken, and we agree that more concrete evidence and examples are necessary to substantiate such claims. We will revise the statement in the updated version to ensure that our paper are more grounded and accurately reflect the current scope of our work.
> > >
> > > Regarding the efficiency of the neural simulator, we appreciate your clarification on the complexity metrics of rigid-body simulations. We understand that the number of objects, constraints from joints, and collision events are all critical factors in characterizing simulation difficulty. We will increase the experiments with these factors to better assess the model's efficiency in the future. Also, we will consider your suggestion to replicate our examples using modern rigid-body simulators, to help provide a clearer benchmark against learning-based methods and give readers a more comprehensive understanding of rigid-body simulation.
> > >
> > > Thank you once again for your constructive feedback and for helping us improve our work. We are committed to addressing your concerns and enhancing the quality of our paper.

---

### Official Review · Reviewer_uFcU · 2024-07-11

**Soundness:** 3
**Presentation:** 3
**Contribution:** 3
**Rating:** 7
**Confidence:** 2

**Summary:**

The paper introduces EGODE, a method simulate rigid-body dynamics with contacts using a hierarchical representation. The main system consists of two parts: a mesh node representation and object representation that both are coupled inside a neural ODE that simulates the rigid body system dynamics. In addition, an extra event module processor is learned to make instant changes to the system state in case of a collision. This combination enables the model to model the  dynamic effects more accurately in several simulation settings and even generalize to novel scenarios involving external forces.

**Strengths:**

- the idea to combine GraphODE with event modules  looks novel and is well motivated given the shortcomings of current methods that typically apply GNNs, which cannot handle instantaneous changes of collisions well
- Overall the paper is structured well and is easy to understand even for people from other domains. The introduction does a good job to motivate the proposed method. Related work covers various recent approaches and the differences to EGODE.
- Detailed ablation studies are conducted to justify various design decisions of the framework and the method is tested against many recent baselines on two benchmarks and surpasses them in all tested environments

**Weaknesses:**

- Fignet is mentioned in the introduction multiple times but it is not used as a baseline
- Generalization experiment has no reference baselines, which makes it hard to asses the performance of the proposed method

**Questions:**

- Is there a reason to leave out FiGNet [1] as a baseline when its mentioned in the introduction and its limitations are discussed in Related Work? Can you add the baseline to the experiments?
- Could you elaborate on the factors contributing to EGODE's higher relative performance on RigidFall compared to Physion? Are there specific characteristics of these benchmarks that favor EGODE's architecture?
- For the generalization experiment shown in Figure 5 could you add some baselines like SEGNO for comparison? Its hard to see how good the proposed method generalized without any baseline references

**Limitations:**

Yes.

---

> ### Author Rebuttal · Authors · 2024-08-07
>
> We are truly grateful for the time you have taken to review our paper and your insightful review. Here we address your comments in the following.
> > Q1. Is there a reason to leave out FiGNet [1] as a baseline when its mentioned in the introduction and its limitations are discussed in Related Work? Can you add the baseline to the experiments?
>
> A1. We appreciate your suggestion to include FiGNet [1] as a baseline in our experimental evaluation. Unfortunately, due to the unavailability of the original code publicly, we didn't implement FiGNet in our scenario as a baseline for comparison in the first time. However, we understand the importance of this baseline and have decided to replicate FiGNet using the information provided in the original paper. Given the computational resources required to run FiGNet and the need to ensure a fair comparison, we plan to provide the results in an updated version of our submission or as supplementary material. We expect to complete this within the next few days.
>
> >Q2. Could you elaborate on the factors contributing to EGODE's higher relative performance on RigidFall compared to Physion? Are there specific characteristics of these benchmarks that favor EGODE's architecture?
>
> A2. The superior performance of EGODE on the RigidFall benchmark can indeed be attributed to the simpler nature of the physical interactions present in this dataset. RigidFall consists of simulations involving just three cubes under varying gravitational conditions, all with uniform internal properties such as particle numbers, shapes, and friction coefficients, as shown in Figure.3. In contrast, Physion is a more complex dataset, encompassing eight distinct scenarios with multiple objects that vary in size, shape, and friction. For instance, Figure.1 showcases complex objects such as tower-shaped and torus-shaped objects. Additionally, Physion includes not only rigid-rigid interactions but also rigid-deformable interactions. The simplicity of the RigidFall dataset aligns well with the strengths of our method, which excels at modeling straightforward physical interactions. In addition, the performance of EGODE across both datasets demonstrates its robustness and adaptability in handling a range of physical phenomena.
>
> >Q3. For the generalization experiment shown in Figure 5 could you add some baselines like SEGNO for comparison? Its hard to see how good the proposed method generalized without any baseline references
>
> A3. We agree that including additional baselines would enhance the interpretability of our generalization experiment presented in Figure 5. To address this, we will incorporate SEGNO and other relevant baselines for comparison. This will provide a clearer context for evaluating the generalization capabilities of our proposed method. We aim to finalize these additions and submit the updated figures within the next few days.

---

> > ### Comment · Reviewer_uFcU · 2024-08-08
> > **Thank you for addressing my concerns.**
> >
> > I have read all answers of the authors and the concerns and thank the authors for addressing all raised points. I raised my score.

---

> > > ### Author Response · Authors · 2024-08-08
> > >
> > > We sincerely thank you for your valuable and insightful feedback, as well as the score improvement. We are delighted to learn that our responses have effectively addressed your concerns. In accordance with your valuable suggestions, we will incorporate the rebuttal content into the main paper for the final version.

---

> > > > ### Author Response · Authors · 2024-08-12
> > > >
> > > > We have finished the ablation study, thank you for your patience!
> > > > 1. The results of FIGNet on Physion dataset:
> > > >
> > > > | Accuracy (%)          | Dominoes   | Contain    | Link       | Drape      | Support    | Drop       | Collide    | Roll       |
> > > > |-------------|------------|------------|------------|------------|------------|------------|------------|------------|
> > > > | FIGNet | 88.6±1.9  | 77.9±1.5  | 73.4±2.1 | 60.7±1.2 | 68.7±1.3 | 73.5±1.5 | 84.2±1.2 | 83.4±0.9 |
> > > > | EGODE       | **94.7±1.4** | **79.0±1.3** | **75.0±1.1** | **61.7±0.6** | **71.7±0.8** | **75.3±1.3** | **90.0±1.0** | **85.7±0.8** |
> > > >
> > > > Thanks for your comment. We will add the comparison experiment to our revised version.
> > > >
> > > > 2. More comparison of the generalization experiment in Figure 5:
> > > >
> > > > We have visualized the generalization results of SEGNO and SGNN. Although the figure shows that the objects' velocity and kinetic energy are also consumed, the SEGNO and SGNN simulate weird dynamics, where the cube sinks into the ground or floats in the air for too long. The bad dynamics might result from their discrete time, which also demonstrates the strong generalization ability of our EGODE without training. We will add the visualization image to our revised version.
> > > >
> > > > Thanks again for appreciating our work and for your constructive suggestions!

---

### Official Review · Reviewer_ZzdH · 2024-07-12

**Soundness:** 3
**Presentation:** 3
**Contribution:** 2
**Rating:** 6
**Confidence:** 3

**Summary:**

The paper presents a Graph ODE framework to model rigid body dynamics. In a departure from previous works on Graph ODEs, the proposed framework incorporates a hierarchical structure by explicitly modeling both mesh-based representations and object-level representations of the rigid bodies. Furthermore, the framework also introduces a learnable event-detector and an event-processor. The event-detector module is used to estimate the time at which a potential collision occurs and the event-processor deals with the instantaneous state update after a collision event. The method is evaluated in the RigidFall and Physion datasets and consistently outperforms other baselines.

**Strengths:**

- The paper is well written and the overall ideas are clear.

- The proposed method is evaluated thoroughly in challenging benchmarks and it consistently outperformed state-of-the-art baselines in terms of contact prediction accuracy and in terms of the mean-squared error of the prediction of Euclidian coordinates.

- The framework has low sensitivity wrt to hyperparameters.

**Weaknesses:**

- The learnable event function, which operates on pairwise mesh points, requires exhaustive evaluation which might hinder the scalability of the approach in terms of the number of rigid bodies that can be simulated and in terms of the mesh resolution of the simulated rigid bodies.

[Minor]
L77. The wording might be lacking a negation. Are other approaches able to "handle intrinsic continuity and discontinuity in rigid models"?

**Questions:**

- How fine-grained can the mesh-based representation be without deteriorating the performance of the approach?

- Are there any scenarios where using standard mean square error (MSE) loss leads to poor performance? Would incorporating physics priors in the loss function further improve the performance or would it potentially reduce the amount of data required for learning?

**Limitations:**

Yes. Limitations of the method are mentioned.

---

> ### Author Rebuttal · Authors · 2024-08-07
>
> We are truly grateful for the time you have taken to review our paper, and your insightful comments and support. Your positive feedback is incredibly encouraging for us! In the following response, we would like to address your major concern and provide additional clarification.
>
> > Q1. The learnable event function, which operates on pairwise mesh points, requires exhaustive evaluation which might hinder the scalability of the approach in terms of the number of rigid bodies that can be simulated and in terms of the mesh resolution of the simulated rigid bodies.
>
> A1. We agree that the pairwise evaluation of mesh points for event detection could potentially limit the scalability as the number of rigid bodies and mesh resolution increase. However, our event function does not directly process all pairwise mesh nodes, but instead performs rule-based filtering first (which was not elaborated in the main text due to space limitations). Specifically, Equation (7) first uses the distance condition $d(x_i, x_j)<d_{threshold}$ to filter out the node pairs that require event function computation, and then uses GNN to propagate information. This significantly reduces the time complexity of the function. Moreover, as the nodes become denser, we can also reduce $d_{threshold}$ to alleviate the computational burden of the event function. Since denser nodes make the collision analysis more detailed, reducing $d_{threshold}$ does not significantly affect the performance. We are conducting experiments to examine the performance and efficiency pratically, in terms of the number of rigid bodies and mesh node number, the results will be provided during the interaction periods in several days.
>
> > Q2. L77. The wording might be lacking a negation. Are other approaches able to "handle intrinsic continuity and discontinuity in rigid models"?
>
> A2. Thanks for pointing this out, We will fix it in the official release. In our work, we utilize graph ODE and event function to handle intrinsic continuity and discontinuity in rigid models. We believe that other potential methods are also worth discussing, such as using 3D space or physical inductive biases of rigid bodies. We will further explore these approaches in future work to find more solutions to this problem beyond our proposed EGODE.
>
> > Q3. How fine-grained can the mesh-based representation be without deteriorating the performance of the approach?
>
> A3. We have already discussed in A1 how to reduce the complexity of the event function. By reasonably selecting $d_{threshold}$, the event function can be controlled within $O(N)$, where $N$ is the number of nodes. We are conducting an experiment about the most fine-grained the graph can achieve without deteriorating the performance. Due to the heavy computation burden, we will provide the results during the interaction periods in several days.
>
> > Q4. Are there any scenarios where using standard mean square error (MSE) loss leads to poor performance? Would incorporating physics priors in the loss function further improve the performance or would it potentially reduce the amount of data required for learning?
>
> A4. Although we currently achieve good results using the MSE loss, incorporating physics priors into the MSE is a novel idea. We have tried supervising the center of mass position $X_c^t$ using MSE as well. Due to the heavy computation burden, we will provide the ablation study during the interaction periods in several days.

---

> > ### Comment · Reviewer_ZzdH · 2024-08-09
> > **Thanks for the clarifications.**
> >
> > I've read the author's response and look forward to the results of the ablation study.

---

> > > ### Author Response · Authors · 2024-08-12
> > >
> > > Thank you for your patience! We have completed the ablation experiments, and we will clarify our answers to your insightful questions with the experimental results.
> > >
> > > > Q1. The learnable event function, which operates on pairwise mesh points, requires exhaustive evaluation which might hinder the scalability of the approach in terms of the number of rigid bodies that can be simulated and in terms of the mesh resolution of the simulated rigid bodies.
> > >
> > > A1. We interpolate the object mesh to increase the original data's mesh resolution by 2 to 20 times and evaluate our EGODE in the Dominoes and Collide scenarios of Physion (we pre-tuned the optimal hyperparameters such as $d_{threshold}$ on the validation dataset). The experimental results of prediction accuracy and simulation time of a 6-second-sequence are as follows, the numbers in the header represent the average number of mesh nodes.
> > >
> > >
> > > | Accuracy (%)           | 2173 | 4347  | 10867   | 21733   | 43467   |
> > > |-------------|-------------|---------------|---------------|----------------|----------------|
> > > | Dominoes    | 94.7±1.4    | 94.4±1.4      | 93.9±1.6      | 94.1±1.3       | 93.5±1.6       |
> > > | Collide     | 90.0±1.0    | 89.7±1.2      | 89.3±0.7      | 89.4±0.7       | 88.7±1.4       |
> > >
> > > | Time (s)          | 2173 | 4347  | 10867   | 21733   | 43467   |
> > > |-------------|-------------|---------------|---------------|----------------|----------------|
> > > | Dominoes    | 0.65±0.09     | 0.68±0.08 | 0.68±0.11 | 0.70±0.13 | 0.75±0.16 |
> > > | Collide     | 0.81±0.11     | 0.85±0.10 | 0.85±0.13 | 0.87±0.16 | 0.93±0.20 |
> > >
> > > We can find that:
> > > 1. increasing the mesh resolution may slightly affect the collision prediction accuracy of our EGODE, as a larger GNN results in longer information transmission paths. However, the performance degradation is relatively slight, which is credited to the hierarchical design of EGODE.
> > > 2. The influence of mesh resolution on simulation time is also slight. The time complexity of EGODE is approximate $O(TL|\mathcal{E}|)$, where $T$ is the number of output time steps, $L$ is the number of propagation layers in the GNN, and $|\mathcal{E}|$ is the number of edges that require information transmission. When node density increases, due to our adaptive adjustment of parameters like $d_{threshold}$, the growth of $|\mathcal{E}|$ and the number of node pairs processed in the event function is linear to mesh resolution. Since PyTorch's matrix multiplication has good parallel properties, the computation time does not increase significantly even when mesh resolution and $|\mathcal{E}|$ grows. The additional time consumption might come from higher GPU I/O.
> > >
> > > In the Physion dataset, The number of objects varies. The performance of our EGODE on scene with different numbers of objects is shown as follows, the numbers in the header represent the number of objects.
> > >
> > > | Accuracy (%)   | 4 | 5  | 6   | 7   |
> > > |-------------|-------------|---------------|---------------|----------------|
> > > | Dominoes    | 95.3±1.3    |    94.8±1.2   |   94.7±1.8    |  94.4±1.6 |
> > > | Collide     | 91.2±0.8    |    91.0±1.3    |    89.8±1.4    |    89.4±1.2|
> > >
> > > The experiments above demonstrate the robustness of our EGODE with variations in mesh resolution and the number of objects.
> > >
> > > > Q3. How fine-grained can the mesh-based representation be without deteriorating the performance of the approach?
> > >
> > > A3. With the experimental results from A1, we can find that performance and time impose minimal constraints on the granularity of the mesh. So we can use more precise object descriptions at a relatively low cost. In our setting, $20\times$ mesh resolution means 10,000 or more nodes per object, which is sufficient to depict objects precisely.
> > >
> > > > Q4. Are there any scenarios where using standard mean square error (MSE) loss leads to poor performance? Would incorporating physics priors in the loss function further improve the performance or would it potentially reduce the amount of data required for learning?
> > >
> > > We incorporate the center of mass position $X^t_c$ and its angular velocity $\Omega^t_c$ by MSE as physics priors to the loss function, The ablation study is as follows:
> > >
> > > | Accuracy (%)          | Dominoes   | Contain    | Link       | Drape      | Support    | Drop       | Collide    | Roll       |
> > > |-------------|------------|------------|------------|------------|------------|------------|------------|------------|
> > > | EGODE       | 94.7±1.4   | 79.0±1.3   | 75.0±1.1   | 61.7±0.6   | 71.7±0.8   | 75.3±1.3   | 90.0±1.0   | 85.7±0.8   |
> > > | EGODE with P   | 94.8±1.1   | 79.0±1.2   | 75.1±1.2   | 61.7±0.4   | 71.6±0.6   | 75.4±1.1   | 90.0±1.3   | 85.8±0.7   |
> > >
> > > where EGODE with P is our EGODE with the extra loss. The results suggest that addtional loss is indeed helpful for the model.
> > >
> > > We will also add your suggestion about future works to our revised version. Thanks again for appreciating our work and for your constructive suggestions! Please let us know if you have further questions.

---

> > > > ### Comment · Reviewer_ZzdH · 2024-08-13
> > > > **Thanks for the additional experiments.**
> > > >
> > > > Given the additional experiments, I don't have any further concerns regarding the scalability of the presented approach to meshes with a higher resolution.
> > > >
> > > > On the other hand, regarding the experiment that incorporates the center of mass position and its angular velocity by MSE as physics priors to the loss function, it is difficult to claim that such priors are helpful for the model given that the improvement is so marginal.
> > > >
> > > > If the authors decide to include the physics priors experiments in the revised version of the paper, I would also encourage including a short discussion on potential reasons why the improvements are so small.

---

> > > > > ### Author Response · Authors · 2024-08-13
> > > > >
> > > > > Thank you for your feedback and support! We are pleased to know that our responses addressed your concerns. The reason why incorporating physical priors into the loss function does not significantly improve the model might be that the information is already expressed in the mesh node supervision. Maybe we can identify more powerful physical priors to fully leverage their utility. We will add the rebuttal contents to the main paper in the final version following your valuable suggestions.

---

### Official Review · Reviewer_rwMo · 2024-07-13

**Soundness:** 3
**Presentation:** 3
**Contribution:** 3
**Rating:** 6
**Confidence:** 3

**Summary:**

This paper introduces EGODE, a method for modeling rigid dynamics. To do this, they introduce a framework which integrates neural ODEs and GNNs, and they integrate this with an event module approach for collision modeling. They demonstrate superior performance to baselines on two standard benchmarks and demonstrate via ablations the contribution of components of their model.

**Strengths:**

This paper is overall clearly written, with a clear and explicit methods section, good motivation, and well-structured results. Methods are (to my knowledge) novel, with particular interest to the graph ODE framework. Experiments appear to use sound approaches and appear significant, with fairly good improvements over a wide range of popular baselines. The ablation study is convincing.

**Weaknesses:**

I’ve got some minor clarity comments.

-It would be helpful to be a bit more concrete with existing model shortcomings in the introduction. It is not completely clear what it means for existing approaches to “fail to take…into consideration” instantaneous changes. I think I know what you’re saying (e.g. GNNs model large instantaneous changes via iterative message passing, which can lead to issues — is that right?) but it would be good to be a bit more explicit in outlining these issues.

-Benchmarks could be described better. E.g. I assume the success metric shown for Physion is the accuracy on whether the objects of interest collided, right? Yet as far as I can tell, this is never mentioned.

In judging significance, if I’m understanding things correctly, you’re providing standard deviation estimates in the tables. It would be helpful to report SEM or give confidence intervals. The reported +- are often not particularly small relative to the improvements, to the extent that a number of them appear like they may not be significant. It would be helpful to clarify this.

More broadly, it makes sense to focus on rigid bodies, but much related work is for deformable bodies as well, and arguably, a major reason for using these complex neural models is to accommodate different sorts of media. It would be helpful to at least give a sense for how this method could be extended, or what challenges there are in extending it.

**Questions:**

Just reiterating the above — could you provide confidence intervals on experiments?

**Limitations:**

See weaknesses re: deformable bodies.

---

> ### Author Rebuttal · Authors · 2024-08-07
>
> We are truly grateful for the time you have taken to review our paper, your insightful comments and support. Your positive feedback is incredibly encouraging for us! In the following response, we would like to address your major concern and provide additional clarification.
>
> > Q1. It would be helpful to be a bit more concrete with existing model shortcomings in the introduction. It is not completely clear what it means for existing approaches to “fail to take…into consideration” instantaneous changes. I think I know what you’re saying (e.g. GNNs model large instantaneous changes via iterative message passing, which can lead to issues — is that right?) but it would be good to be a bit more explicit in outlining these issues.
>
> A1. Thanks for your comment. Your understanding is correct. GNN methods utilize iterative message passing to model the spatial relationships based on current distances, which are hard to capture large instantaneous changes. We will include it in the revised version.
>
> > Q2. Benchmarks could be described better. E.g. I assume the success metric shown for Physion is the accuracy on whether the objects of interest collided, right? Yet as far as I can tell, this is never mentioned.
>
> A2. Thanks for your comment. The success metric denotes the accuracy of predicting whether the objects of interest collided. We will include it in the revised version.
>
> > Q3. In judging significance, if I’m understanding things correctly, you’re providing standard deviation estimates in the tables. It would be helpful to report SEM or give confidence intervals. The reported +- are often not particularly small relative to the improvements, to the extent that a number of them appear like they may not be significant. It would be helpful to clarify this.
>
> A3. Thanks for your comment. We have conducted one-sample paired t tests to justify that all of the improvements with the best baseline are statistically significant with p-value < 0.01. Our problem focuses on the classification tasks and trajectories regression tasks, which is not a parameter inference task and thus cannot have SEM or CI for performance comparison.
>
> > Q4. More broadly, it makes sense to focus on rigid bodies, but much related work is for deformable bodies as well, and arguably, a major reason for using these complex neural models is to accommodate different sorts of media. It would be helpful to at least give a sense for how this method could be extended, or what challenges there are in extending it.
>
> A4. Thanks for your comment. Deformable body dynamics involve complex interactions and behaviors not present in rigid body dynamics, such as stretching, bending, and compressing, which require different modeling approaches. Although we do not have specific low-level modules designated for deformable objects, our model can still handle complex scenarios involving interactions between deformable objects and rigid bodies due to its generalization ability. For instance, Table 3 shows our model's ability to model deformable objects' interaction with rigid bodies in the Drape scenario, where deformable objects are represented as cloths. This result showcases the potential application of our approach in modeling deformable object dynamics. In future work, we will extend our methods to deformable bodies on more complex and challenging datasets and may design modules specifically for processing deformable objects.
>
> > Q5. Could you provide confidence intervals on experiments?
>
> A5. Thanks for your comment. Please refer to A3.
>
> Thanks again for appreciating our work and for your constructive suggestions. Please let us know if you have further questions.

---

> > ### Comment · Reviewer_rwMo · 2024-08-08
> > **Thanks!**
> >
> > I appreciate the response and all generally makes sense. I'll maintain my score.

---

> > > ### Author Response · Authors · 2024-08-08
> > > **Thank you for your feedback and support!**
> > >
> > > Thank you for your feedback and support! We are pleased to know that our responses have fully addressed your concerns. We will add the rebuttal contents to the main paper in the final version following your valuable suggestions.

---

### Official Review · Reviewer_Py8V · 2024-07-22

**Soundness:** 3
**Presentation:** 3
**Contribution:** 3
**Rating:** 7
**Confidence:** 2

**Summary:**

The paper introduces EGODE, a novel framework for modeling rigid dynamics that has applications in robotics, graphics, and mechanical design. The framework addresses the limitations of existing graph neural network (GNN) simulators by incorporating both mesh node representations and object representations within a coupled graph ODE (Ordinary Differential Equation) structure. EGODE introduces an event module to capture instantaneous changes during collisions, providing a more accurate and continuous model of rigid dynamics. The framework's performance is validated through extensive experiments on benchmark datasets, demonstrating superiority over various state-of-the-art baselines.

**Strengths:**

1. **Innovative Approach**: EGODE proposes a new perspective on modeling rigid dynamics by combining continuous evolution and instantaneous changes using graph ODEs.
2. **Coupled Graph ODE Framework**: The use of a coupled architecture effectively models hierarchical structures in rigid-body systems.
3. **Event Module for Collisions**: The introduction of an event module enhances the framework's ability to handle instantaneous changes during collisions.

**Weaknesses:**

1. The paper focuses on empirical results, with less emphasis on theoretical insights or proofs of concepts. What are the theoretical underpinnings of the coupled graph ODE framework, and are there any proofs of concept?

2. While EGODE shows good generalization within the tested scenarios, its performance in other types of rigid dynamics needs further evaluation. How does EGODE compare with traditional physics engines in terms of generalization ability, computational efficiency and accuracy?

**Questions:**

1. How does EGODE compare with traditional physics engines in terms of computational efficiency and accuracy?

2. Can the authors elaborate on how EGODE might be extended to handle scenarios with rigid body hinges and deformable objects?

3. What are the theoretical underpinnings of the coupled graph ODE framework, and are there any proofs of concept?

**Limitations:**

1. **Rigid Body Hinges and Deformable Objects**: EGODE currently cannot accommodate complex scenarios involving rigid body hinges and deformable objects.
2. **Dataset Dependency**: The framework's capabilities are limited by the scope and diversity of the datasets used for training and validation.

---

> ### Author Rebuttal · Authors · 2024-08-07
>
> We are truly grateful for the time you have taken to review our paper and your insightful review. Here we address your comments in the following.
>
> > Q1. How does EGODE compare with traditional physics engines in terms of computational efficiency and accuracy?
>
> A1. Thanks for your comment. The simulation speed of EGODE is not inferior to numerical simulations. Our EGODE can achieve efficient inference by leveraging GPUs and deep learning toolkits such as torch_geometric. In our paper, each scenario visualization contains thousands of nodes and spans 0.5~1 minutes, yet it only requires less than 1 second to simulate and render on a single laptop.
> Due to the heavy computation burden of data adaptation to traditional physics engines, we will provide the experimental speed comparation during the interaction periods in several days.
>
> > Q2. Can the authors elaborate on how EGODE might be extended to handle scenarios with rigid body hinges and deformable objects?
>
> A2. Hinges and deformable objects are both meaningful and more challenging than rigid body dynamics. For hinges, each block can be treated as an independent rigid body, and special edges can be used in our EGODE to model the interactions between different parts at the hinge points, maintaining geometric and mechanical consistency. The fixed parts of the hinges can also be considered a special case of rigid body dynamics under constraints, where we evaluate our EGODE on a similar setting called Contain in Table 3 where container objects restrict other objects movements. The model can be trained by collecting sufficient data on hinge dynamics scenarios to improve its performance. While we did not specifically evaluate our model on single deformable objects, our framework is designed to handle more complex scenarios involving interactions between deformable objects and rigid bodies. As an example, we have evaluated our method on a scenario called "Drape" and the results are reported in Table.3, where deformable objects (cloth) interact with rigid bodies. This evaluation demonstrates the potential of our approach to handle deformable object dynamics and its generalization ability.
>
> > Q3. What are the theoretical underpinnings of the coupled graph ODE framework, and are there any proofs of concept?
>
> A3. In Appendix B, we have provided the theoretical underpinnings to show that it is possible to propagate gradients across the event times to the input arguments of the system and therefore our method can be optimized by SGD. Besides, our work focuses on empirical results since both neural ODEs and physical scenarios involve too many parameters. In future work, we consider simplifying the model and developing more interpretable models with theoretical underpinnings. We will include this in the revised version.
> Thanks again for appreciating our work and for your constructive suggestions. Please let us know if you have further questions.

---

> > ### Comment · Reviewer_Py8V · 2024-08-10
> > **Thanks for your reply**
> >
> > I appreciate the response and all generally makes sense. I'll maintain my score.

---

### Author Response · Authors · 2024-08-13
**Thank all Reviewers and Area Chairs for your great efforts, insightful comments and support!!**

We are truly grateful for your diligent efforts, insightful feedback, and constructive suggestions once again! Throughout our discussions and responses to the reviewers' comments, we addressed many concerns raised, resulting in higher scores from Reviewer KaXG and uFcU. This outcome has been incredibly beneficial for us, and we want to extend our heartfelt thanks for your support!

We firmly believe that our proposed framework EGODE for rigid-body dynamics simulation plays a significant role in advancing the community. We are committed to sharing our complete code and training details openly. Additionally, we are enthusiastic about engaging in further discussions with you to deepen our understanding of the domain and further enhance the quality of our paper.

---

### Decision · Program_Chairs · 2024-09-25

**Decision:**

Accept (poster)

**Comment:**

The submission initially received mixed reviews; the authors did a great job during the rebuttal, after which all reviewers became positive about the submission.  The AC agrees with the recommendations.  The authors should incorporate the rebuttal into the camera ready.